# Randomized controlled trial for time-restricted eating in healthy volunteers without obesity

Zhibo Xie[1,6], Yuning Sun[2,6], Yuqian Ye[2,3,6], Dandan Hu[1,4], Hua Zhang[5], Zhangyuting He[2], Haitao Zhao[1], Huayu Yang [1✉] & Yilei Mao [1✉]

Time-restricted feeding (TRF) improves metabolic health. Both early TRF (eTRF, food intake restricted to the early part of the day) and mid-day TRF (mTRF, food intake restricted to the middle of the day) have been shown to have metabolic benefits. However, the two regimens have yet to be thoroughly compared. We conducted a five-week randomized trial to compare the effects of the two TRF regimens in healthy individuals without obesity (ChiCTR2000029797). The trial has completed. Ninety participants were randomized to eTRF (n=30), mTRF (n=30), or control groups (n=30) using a computer-based random-number generator. Eighty-two participants completed the entire five-week trial and were analyzed (28 in eTRF, 26 in mTRF, 28 in control groups). The primary outcome was the change in insulin resistance. Researchers who assessed the outcomes were blinded to group assignment, but participants and care givers were not. Here we show that eTRF was more effective than mTRF at improving insulin sensitivity. Furthermore, eTRF, but not mTRF, improved fasting glucose, reduced total body mass and adiposity, ameliorated inflammation, and increased gut microbial diversity. No serious adverse events were reported during the trial. In conclusion, eTRF showed greater benefits for insulin resistance and related metabolic parameters compared with mTRF. Clinical Trial Registration URL: http://www.chictr.org.cn/showproj.aspx?proj=49406.

[1] Department of Liver Surgery, Peking Union Medical College (PUMC) Hospital, PUMC & Chinese Academy of Medical Sciences, Beijing, China. [2] Peking Union Medical College (PUMC), PUMC & Chinese Academy of Medical Sciences, Beijing, China. [3] Department of Head and Neck Surgery, National Cancer Center/National Clinical Research Center for Cancer/Cancer Hospital, Chinese Academy of Medical Sciences and Peking Union Medical College, Beijing, China. [4] Department of Hepatobiliary Surgery, Sun Yat-sen University Cancer Center, Guangzhou, Guangdong, China. [5] Department of Vascular Surgery, Peking Union Medical College (PUMC) Hospital, PUMC & Chinese Academy of Medical Sciences, Beijing, China. [6] These authors contributed equally: Zhibo Xie, Yuning Sun, Yuqian Ye. ✉email: dolphinyahy@hotmail.com; pumch-liver@hotmail.com

Long-term dietary habits are determinants of metabolic health[1]. In particular, western-style diets, which include fat-rich food and snacks that are consumed around the clock, play a causative role in the development of some chronic diseases[2]. In contrast, our human ancestors did not have continuous access to a food supply. There are also a number of more modern customs that involve fasting, such as "Sustenance" in China and Ramadan for Muslims, which demonstrate that fasting is possible in daily life. Time-restricted feeding (TRF), in which the daily window for food consumption is restricted to a period of 4–10 h[3], is thought to be a feasible fasting regimen for most people. It has previously been shown in animal models that TRF has many beneficial effects, including a reduction in body mass, the prevention of obesity, an improvement in insulin sensitivity, a reduction in hepatic fat content, the prevention of hepatosteatosis and hyperlipidemia, and the amelioration of hepatic ischemia-reperfusion injury and inflammation[4–8]. TRF has also been shown to induce weight loss, increase insulin sensitivity, and reduce blood pressure, and to have other metabolic benefits in humans[3,9–18]. Furthermore, no serious side-effects have been documented during previous studies of TRF in humans[19,20]. Previous studies have shown that the effects of TRF might depend on the particular time window for food consumption, but the exact time periods used to date have not been well defined. Nevertheless, restricting food intake to the early part of the day (early TRF; eTRF) or to the middle part of the day (mid-day TRF; mTRF) have been studied[3,10,13]. On the other hand, late day TRFs were shown to produce inconsistent effects, with a recent trial showing reductions in body weight, insulin resistance, and oxidative stress after 4 h late day TRF[21], while some others does not produce the same magnitude of improvements as either eTRF or mTRF[15,22,23]. Daily rhythm variations have been suggested to be associated with the differing effects of TRFs during different periods of the day in animal studies[24,25], and in humans[26].

We performed a randomized controlled trial to compare the effects of eTRF (eating during a period of no more than 8 h between 06:00 and 15:00, and fasting for the rest of the day), mTRF (eating during a period of no more than 8 h between 11:00 and 20:00, and fasting for the rest of the day) with a control group (eating ad libitum) for 5 weeks on health-related parameters in persons without obesity. We also evaluated the daily rhythms of plasma adipokine concentrations and clock gene expression in peripheral blood mononuclear cells (PBMCs). We found that eTRF was associated with a larger improvement in insulin sensitivity than mTRF, and eTRF, but not mTRF, was associated with lower fasting plasma glucose, body mass, adiposity, and inflammation; and a more diverse gut microbiota than the control group. In addition, the two types of TRF had differing effects on the daily rhythms of plasma adipokines and PBMC clock gene expression. Taken together, these findings suggest that TRF with a window for food consumption early in the day is better for metabolic health than a window later in the day, and that the mechanism may involve changes to daily rhythms.

## Results

**Participants**. Ninety volunteers who met the eligibility criteria participated in the trial and were randomized at a ratio of 1:1:1 to eTRF, mTRF, and control groups (Fig. 1). Eighty-two participants (91.1%) completed the entire five-week trial; two, four, and two participants in the eTRF, mTRF, and control groups did not. Self-reported compliance with the regimens was 949 out of 980 person-days (96.8%) in the eTRF group, and 894 out of 910 person-days (98.2%) in the mTRF group. Comparisons of the groups with respect to age, sex distribution, body mass, and body mass index (BMI) at baseline are shown in Table 1. No serious adverse events were reported during the trial.

**Energy intake**. Energy intake was estimated using pictures of the meals consumed. There was a significant difference in the change in energy intake among the eTRF, mTRF, and control groups ($p < 0.001$), with the eTRF ($\Delta = -240 \pm 409$ kcal/day) and mTRF ($\Delta = -159 \pm 397$ kcal/day) groups showing a reduction in energy intake, whereas the control group ($\Delta = 64 \pm 286$ kcal/day; $p < 0.001$ and $p < 0.01$, respectively) did not. However, there was no difference between the two TRF groups ($p = 0.30$, Fig. 2a).

**Insulin resistance and fasting glucose**. There were significantly different changes in the Homeostatic Model Assessment of Insulin Resistance (HOMA-IR) among the groups (ANOVA $p < 0.001$, Fig. 2b)[27]. The eTRF group showed a larger reduction in HOMA-IR ($\Delta = -1.08 \pm 1.59$) than either the mTRF group ($\Delta = 0.39 \pm 0.71$, $p < 0.001$) or the control group ($\Delta = -0.05 \pm 0.75$, $p = 0.002$), but there was no difference between the changes in the mTRF and control groups. The change in FPG significantly differed across the three groups (ANOVA $p = 0.007$, Fig. 2c), which reflected a significant difference between the eTRF group ($\Delta = -0.59 \pm 0.84$ mmol/L) and control group ($\Delta = 0.16 \pm 0.38$ mmol/L, $p = 0.005$). There were no differences in the change in FPG between the mTRF ($\Delta = -0.18 \pm 1.17$ mmol/L) and control groups or between the TRF groups.

**Body mass and composition**. Multiple comparisons following ANOVA showed that there was a greater reduction in body mass in the eTRF group ($\Delta = -1.6 \pm 1.4$ kg) than in the control group ($\Delta = 0.3 \pm 1.2$ kg, $p = 0.009$), but there were no differences in the changes in body mass between the mTRF ($\Delta = -0.2 \pm 2.2$ kg) and control groups, or between the TRF groups (Fig. 2d). In addition, the eTRF group showed significantly larger reductions in percentage body fat and body fat mass ($\Delta = -0.60 \pm 1.22\%$ and $\Delta = -0.76 \pm 1.01$ kg, respectively) than the control group ($\Delta = 0.42 \pm 1.16\%$ and $\Delta = 0.41 \pm 0.89$ kg, respectively; $p = 0.042$ and 0.001, respectively). No differences in these two parameters were found between the mTRF ($\Delta = -0.22 \pm 1.70\%$ and $\Delta = -0.30 \pm 1.25$ kg, respectively) and control groups or between the TRF groups (Fig. 2e, f).

**Blood pressure and lipid concentrations**. The changes in systolic blood pressure (SBP), diastolic blood pressure (DBP), and mean arterial pressure (MAP) did not significantly differ among the eTRF, mTRF, and control groups ($p = 0.078$, $p = 0.099$, and $p = 0.088$, respectively). There were also no differences found in the circulating high-density lipoprotein-cholesterol (HDL-C) ($p = 0.28$), low-density lipoprotein-cholesterol (LDL-C) ($p = 0.68$), total cholesterol ($p = 0.94$), and triglyceride ($p = 0.71$) concentrations (Table S1).

**Inflammatory markers, liver enzymes, and immune cells**. Tumor necrosis factor-α (TNF-α), interleukin-8 (IL-8), and high sensitivity C-reactive protein (hsCRP) are important circulating markers of inflammation[28–30]. The change in TNF-α was larger in the eTRF group ($\Delta = -0.81 \pm 1.98$ pg/mL) than in the control group ($\Delta = 0.39 \pm 1.35$ pg/mL, $p = 0.024$), but there were no differences between the mTRF group ($\Delta = -0.06 \pm 0.95$ pg/mL) and the control group, or between the two TRF groups (Fig. 2g). Similarly, there was a larger reduction in the IL-8 concentration in the eTRF group ($\Delta = -1.9 \pm 4.5$ pg/mL) than in the control group ($\Delta = 1.1 \pm 2.9$ pg/mL, $p = 0.045$), but there were no differences between the mTRF group ($\Delta = -1.0 \pm 5.5$ pg/mL) and the control group, or between the TRF groups (Fig. 2h). However, the changes in hsCRP were similar in the three groups ($p = 0.70$, Table S1). The changes in plasma aspartate aminotransferase (AST) activity significantly differed among the three groups (eTRF:

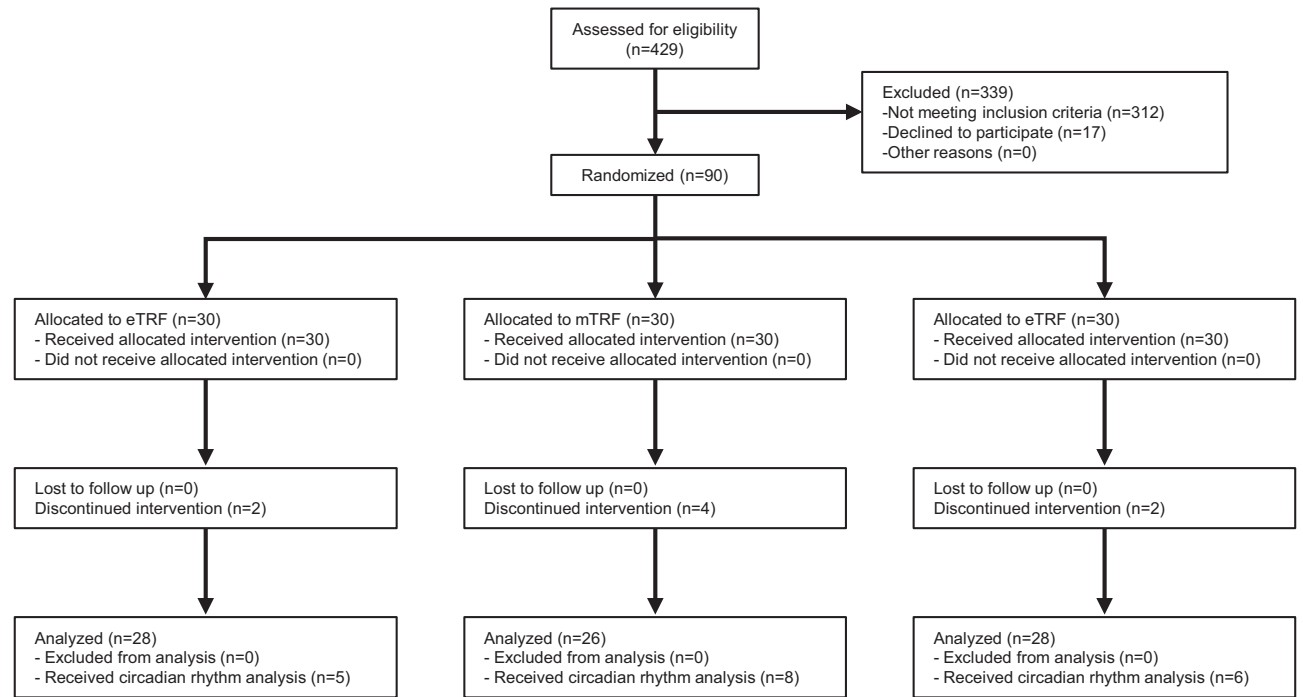

**Fig. 1 CONSORT Flow Diagram.** A total of 429 individuals were assessed for eligibility and 312 were excluded as they did not meet inclusion criteria, and 17 were excluded as they declined to participate. Ninety participants were randomized into one of three groups: eTRF, mTRF, or control, and baseline measures were assessed after randomization. During the 5-week intervention, eight participants (eTRF: $n = 2$; mTRF: $n = 4$; control: $n = 2$) dropped out due to dissatisfaction with the randomization result or for personal scheduling conflicts. Remaining 82 participants completed the trial and were included in the present analysis.

**Table 1 Baseline Characteristics.**

| Characteristics | eTRF | mTRF | control | *p* value |
|---|---|---|---|---|
| No. | 28 | 26 | 28 | |
| Age, mean (SD), years | 28.68 (9.707) | 31.08 (8.438) | 33.57 (11.6) | 0.16 |
| Female | 24 (85.7%) | 19 (73.1%) | 21 (75.0%) | |
| Weight, mean (SD), kg | 61.1 (8.8) | 61.0 (11.7) | 61.2 (9.9) | 0.99 |
| BMI, mean (SD) | 22.7 (3.1) | 21.4 (2.2) | 21.5 (2.9) | 0.68 |

One-way repeated measures ANOVA followed by Holm-Sidak's multiple comparisons test for between group comparisons.
*BMI* body mass index (calculated as weight in kilograms divided by height in meters squared), *eTRF* early time-restricted feeding group, *mTRF* mid-day time-restricted feeding group.

$\Delta = -3.0 \pm 7.0$ U/L, mTRF: $\Delta = -1.0 \pm 4.2$ U/L, control: $\Delta = 0.4 \pm 2.8$ U/L; $p = 0.046$, Fig. 2i), which reflected a larger decrease in the eTRF group than in the control group ($p = 0.041$). However, the activities of alanine aminotransferase (ALT), alkaline phosphatase (ALP), and gamma-glutamyltransferase (GGT) did not differ among the three groups (Table S1).

There were no differences in the changes in white blood cell (WBC) count among the three groups. The eTRF group ($\Delta = 0.8 \pm 1.6\%$) showed a larger increase in the percentage of peripheral T-regulatory cells (pTregs) than the control group ($\Delta = -0.2 \pm 0.9\%$; $p = 0.038$), but there were no differences in this parameter between the mTRF ($\Delta = 0.7\%$, $-0.5$ to $1.4\%$) and control groups or between the two TRF groups (Fig. 2j).

**Gut microbiota.** The changes in the gut microbial α-diversity significantly differed among the eTRF ($\Delta = 18.0 \pm 44.0$), mTRF ($\Delta = 11.2 \pm 51.6$), and control ($\Delta = -10.2 \pm 35.0$) groups ($p = 0.049$, Fig. 2k), with the eTRF group experiencing a larger increase in α-diversity than the control group ($p = 0.048$)[31]. However, there were no significant differences between the mTRF

and control groups ($p = 0.18$) or between the TRF groups ($p = 0.84$).

LEfSe analysis was used to identify bacterial taxa that differed in abundance between the beginning and end of the study in each group. No significant differences were found in the relative abundances of taxa between these time points. However, in the mTRF group, the relative abundances of the genera *Escherichia/Shigella* and *Weissella* were high at baseline, and that of the family *Leuconostocaceae* was high at the end of the study (Fig. S3).

Phylogenetic Investigation of Communities by Reconstruction of Unobserved States (PICRUSt) was used to analyze the representation of gene functions in the microbial communities in each group. We predicted function using clusters of orthologous group (COG) analysis, and found 29, 26, and 1 significantly different functional COGs between baseline and follow-up testing in the eTRF group, mTRF group, and control group, respectively (Fig. S4).

**Sleep quality and appetite.** The Pittsburgh Sleep Quality Index (PSQI) was used to evaluate the sleep quality of the participants; a higher PSQI score is indicative of worse sleep quality[32]. The

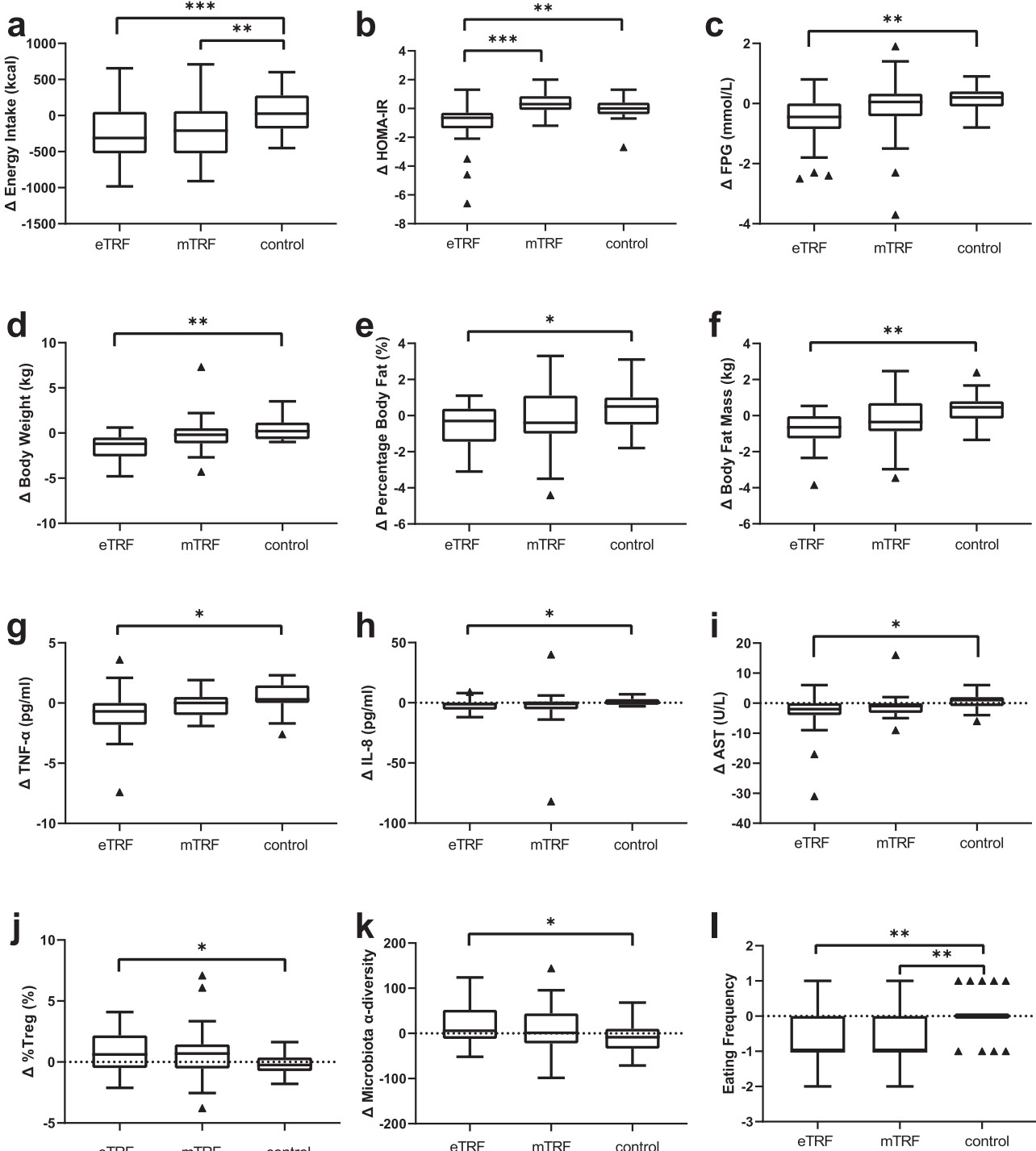

**Fig. 2 Energy intake and metabolic health-related parameters. a** Change in daily energy intake after 5 weeks of intervention, ***$p < 0.001$, **$p = 0.009$. **b** Change in insulin resistance index (measured with HOMA-IR) after 5 weeks of intervention, ***$p < 0.001$, **$p = 0.002$. **c** Change in fast plasma glucose (FPG) after 5 weeks of intervention, **$p = 0.005$; **d** Change in body mass after 5 weeks of intervention, **$p = 0.009$. **e** Change in percentage body fat after 5 weeks of intervention, *$p = 0.042$. **f** Change in body fat mass after 5 weeks of intervention, **$p = 0.001$. **g** Change in TNF-α (tumor necrosis factor-α) after 5 weeks of intervention, *$p = 0.024$. **h** Change in IL-8 (interleukin-8) after 5 weeks of intervention, *$p = 0.045$. **i** Change in AST (aspartate transaminase) after 5 weeks of intervention, *$p = 0.046$. **j** Change in plasma Tregs (T regulatory cells) after 5 weeks of intervention, *$p = 0.038$. **k** Change in gut microbiota α-diversity after 5 weeks of intervention, *$p = 0.048$. **l** Change in eating frequency after 5 weeks of intervention, **$p = 0.001$. $n = 28$ participants in eTRF group, $n = 26$ participants in mTRF group, $n = 28$ participants in control group. Data is visualized as Tukey box plots (line at mean, top of the box at the 75th percentile, bottom of the box at the 25th percentile, whiskers at the highest and lowest values, outliers shown as triangles beyond the whiskers). *$p < 0.05$, **$p < 0.01$, ***$p < 0.001$. One-way repeated measures ANOVA followed by Holm-Sidak's multiple comparisons test for between group comparisons.

changes in PSQI did not significantly differ among the eTRF ($\Delta = -1.08 \pm 1.78$), mTRF ($\Delta = -0.22 \pm 2.19$), and control ($\Delta = -0.36 \pm 1.73$) groups ($p = 0.24$, Table S1).

The Council of Nutrition Assessment Questionnaire (CNAQ) is used to evaluate the appetite of individuals[33]. There were no significant differences in the CNAQ results among the three groups (eTRF: $\Delta = -0.24 \pm 2.30$, mTRF: $\Delta = -0.83 \pm 2.50$, control: $\Delta = 0.07 \pm 1.82$; $p = 0.35$). However, there were differences in the changes in the frequency of food consumption, including of formal meals and snacks ($p < 0.001$, Fig. 2l) among the eTRF ($\Delta = -0.67 \pm 0.76$), mTRF ($\Delta = -0.72 \pm 0.67$), and control ($\Delta = 0.04 \pm 0.59$) groups, with both TRF groups showing significantly lower meal frequency than the control group (eTRF: $p = 0.001$, mTRF: $p = 0.001$).

**Daily rhythms of plasma adipokine concentrations and PBMC clock gene expression.** The fasting plasma concentrations of resistin, leptin, and ghrelin were measured, and no significant changes were found in any of the three groups with respect to the concentrations of these substances. Furthermore, blood samples were collected at four times of day (07:00, 12:00, 17:00, and 23:00) at both the beginning and end of the interventions from 19 participants (five, eight, and six from the eTRF, mTRF, and control groups, respectively). Two-way ANOVA showed that the plasma ghrelin concentration at 23:00 was significantly increased by eTRF ($0.49 \pm 0.17\%$, $p = 0.037$); and the resistin concentration increased at 12:00 ($0.22\% \pm 0.085\%$, $p = 0.048$) and decreased at 17:00 ($-0.24\% \pm 0.078\%$, $p = 0.03$) in the eTRF group. No changes in the concentrations of any of these substances were identified in the mTRF group (Fig. 3).

The expression of the clock genes *BMAL1 (ARNTL), SIRT1, PER1, PER2, PER3, CRY1*, and *CRY2* was measured in PBMCs of the same 19 participants, and the amplitudes, midline-estimating statistics of rhythm (MESORs), and acrophases of the expression levels of these genes were calculated using Cosinor analysis. Because not all the expression levels of the clock genes could be fitted using Cosinor curves, the parameters were calculated for exploratory purposes only (plots for each participant are shown in Fig. S5 and the corresponding $r^2$ values in Table S2). In the eTRF group, the *SIRT1* mRNA expression of all the participants increased in amplitude during the trial (Fig. 4a). Although participants in the mTRF group all showed increases in the amplitude of *PER2* mRNA expression, they also showed decreases in the amplitude of *PER1* mRNA expression (Fig. 4b). In addition, all the participants in the eTRF group showed increases in the MESORs for *BMAL1, PER2*, and *SIRT1* mRNA expression during the trial (Fig. S1a). All the participants in the mTRF group also showed an increase in the MESOR for *PER2* mRNA expression, but the decreases in the MESOR for *PER1* mRNA expression were quite consistent in this group (Fig. S1b). No consistent shifts in the acrophases of the expression of clock genes were identified in any of the groups (Fig. S2).

## Discussion

The present study has shown that 5 weeks of eTRF, but not mTRF, improves insulin sensitivity, reduces fasting plasma glucose, reduces body mass and adiposity, ameliorates inflammation, and increases gut microbial diversity. However, there were no significant differences among the three groups with respect to blood pressure, circulating lipid concentrations, HbA1c, hsCRP, sleep quality, or appetite.

The good compliance with the two TRF protocols in the present study implies that TRF is an easy-to-execute fasting regimen, and the similar compliance with each suggests that they are similarly feasible. The participants in both TRF groups were instructed to eat ad libitum during their daily 8 h eating periods and no specific nutritional guidance was given to them, such that the trial conditions were similar to real-life situations. However, there were reductions in energy intake in both of the TRF groups, which implies that energy intake can be limited just by shortening the daily duration of food consumption. Furthermore, the lack of a significant difference in the change in energy intake between the two TRF groups suggests that the differences in the improvements in metabolic health were not caused by differences in energy intake.

The benefits of improving insulin sensitivity are numerous[34]. Consistent with the results of previous studies[3,14,35], we found that eTRF, but not mTRF, improved insulin sensitivity. Remarkably, this is the first trial to show that eTRF is superior to mTRF with respect to its ability to improve insulin sensitivity by directly comparing these two TRF regimens.

Although similar changes in energy intake occurred in both TRF groups, only the eTRF group showed a reduction in body mass *versus* the control group, which was accompanied by reductions in both the percentage body fat and fat mass. These may indicate an improvement in fat deposition, which requires further visceral fat measuring parameters in future trials. Besides, the weight loss in eTRF group was relatively modest compared with prior eTRF studies[3,36], which may be the result of different inclusion criteria of participants, with normal weight humans included in this trial, while mostly overweight participants or individuals with obesity were included in prior eTRF studies[3,36].

Only one trial by Courtney Peterson et al. has previously reported the effect of eTRF on blood pressure, with participants showing markedly reduced blood pressure after eTRF[3]. In contrast, no significant changes in blood pressure were noticed in the eTRF group in the present trial. The baseline blood pressure levels might be the reason for the different effects of eTRF between these two trials, because the trial by Courtney Peterson et al. was carried on those with a mean blood pressure within pre-hypertensive range[3], while the present trial on healthy participants. The reported effects of mTRF on blood pressure have been inconsistent, and only one previous study showed a reduction on blood pressure, which was assumed to be an "add-on" effects of anti-hypertensive drugs[14,21,26]. In addition, there was no effect of either TRF regimen on circulating lipid concentrations, but this was not unexpected, because the blood concentrations of most of the participants were within the normal range.

Excess nutrient intake usually induces an inflammatory response, which has been causally linked to the dysregulation of glucose and lipid metabolism[37]. Previous studies have shown beneficial effects of TRF to reduce inflammation in individuals with obesity or metabolic diseases[3,22,38], and we have shown that eTRF reduces inflammation in individuals without obesity, in the form of reductions in the plasma concentrations of TNF-α and IL-8. A high plasma AST activity is a feature of obesity-induced hepatic steatosis[39,40], and we have also shown a potential protective effect of eTRF against high liver enzyme activity, which is consistent with the results of most previous studies of animal models of liver steatosis, non-alcoholic fatty liver disease[4,6,41–47], or hepatic ischemia-reperfusion[5]; only one previous study showed that TRF does not affect the activities of the liver enzymes ALT, ALP, and GGT[48].

The increase in pTregs in the eTRF group may also contribute to the beneficial effects of eTRF on metabolism, because a low pTreg count is associated with obesity, insulin resistance, and inflammatory responses[49–53]. Although the mechanism of the effect of TRF on pTregs is still under investigation, intermittent fasting has been shown to increase the number of pTregs in rodent intestines, where they have an immunoregulatory effect[54]. We also found that the α-diversity of the gut microbiota increased

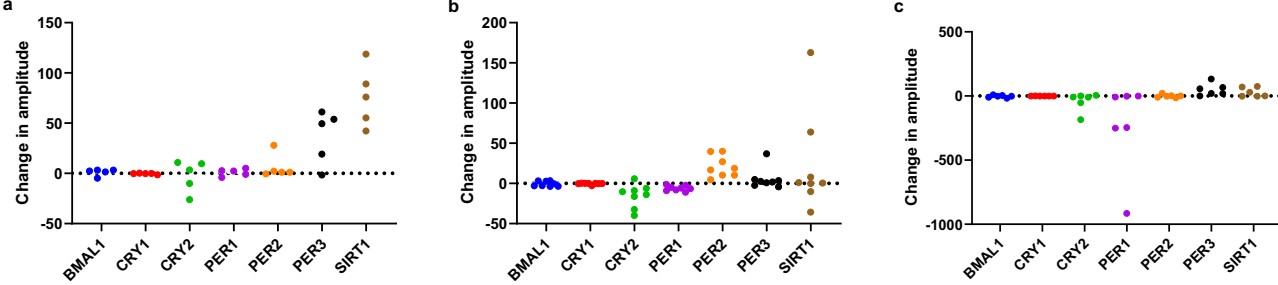

**Fig. 3 Only eTRF, but not mTRF, influenced the daily rhythms of plasma adipokine concentrations.** At baseline and at the end of the study, samples were collected from 19 participants (eTRF, $n = 5$; mTRF, $n = 8$; and control, $n = 6$). Plasma adipokine concentrations (resistin, leptin, and ghrelin) were measured at four time points. Two-way ANOVA analysis showed that eTRF (**a**, **d**, **g**) caused significant changes in the resistin concentration at 12:00 ($p = 0.048$) and 17:00 ($p = 0.03$) (**a**), and in the ghrelin concentration at 23:00 ($p = 0.037$) (**g**), but mTRF (**b**, **e**, **h**), or control groups (**c**, **f**, **i**) had no significant effects on the rhythm of adipokines. The values at each time point are displayed as the percentage of the mean value across all the time points and are displayed as means ± SEMs. Two-way repeated-measures ANOVA with time of day and feeding regimens as the two independent variables. ξ$p < 0.05$ between baseline and follow-up at that time point. BL baseline, FU follow up.

**Fig. 4 TRF influences the daily rhythm amplitude of clock genes expression in peripheral blood mononuclear cells.** After Cosinor analysis, the amplitudes of clock gene expression in each individual were calculated. **a** The change in amplitude of clock genes after analyzed with Cosinor analysis in eTRF group. All participants in eTRF group showed an increase in the amplitude of SIRT1 after the trial. **b** The change in amplitude of clock genes after analyzed with Cosinor analysis in mTRF group. All participants in mTRF group showed an increase in the amplitude of PER2, but a decrease in that of PER1 after the trial. **c** The change in amplitude of clock genes after analyzed with Cosinor analysis in control group.

in the eTRF group, and this has been reported to be associated with a healthier gut microbiota[55], whereas low gut microbial diversity is associated with metabolic diseases[56].

Our finding that eTRF has superior effects to mTRF may be the results of different effects on mediators of the peripheral daily rhythm. Disturbances in the daily rhythms of secreted substances are associated with obesity and metabolic health[57–59], and diet

influences these[60–63]. A previous study conducted in rodents showed that food restriction within an active period influences the daily rhythm of these substances and improves metabolic health[64]. Although in the present study TRFs seemed to have no effects on the fasting plasma concentrations of these substances, eTRF influenced the daily rhythms of ghrelin and resistin. Although the daily variations in the circulating concentration of

resistin has been reported to be related to feeding rhythm in rats[65], and TRF has been reported to influence the circulating concentration of resistin in men[66], the effect of a change in the feeding window on the daily rhythm of resistin had not been previously reported. This change might merely be a reaction to the change in feeding rhythm. The daily rhythm in circulating ghrelin concentration has been reported to synchronize with TRF in mice, with the concentration increasing before the feeding period[67], and it should be noted that ghrelin has an important role in the feeling of hunger[68]. Therefore, the higher concentration of ghrelin that was identified at 23:00 in the eTRF group might be at least in part the results of a longer period of fasting in this group at that time point.

Cosinor analysis has been reported to accurately reflect the rhythmic changes in clock gene expression[69]; therefore, we used this to compare clock gene expression among the groups. Because not all the expression data for every participant fitted Cosinor curves, the parameters were calculated just for exploratory purposes, to provide clues for future investigations. Previous studies identified a positive relationship between the amplitude of oscillation of rhythmic components and metabolic health[70]. We found that eTRF might enhance the daily rhythms in human clock genes, on the basis of the findings that all the participants in the eTRF group showed increases in the amplitude of *SIRT1* expression and the MESORs of *BMAL1*, *PER2*, and *SIRT1* expressions in PBMCs. In contrast, mTRF had diverse effects on the daily rhythms of expression of several clock genes: it increased the amplitude and MESOR of *PER2* expression, but reduced the amplitude and MESOR of *PER1* expression in all the participants. This suggests that mTRF has relatively complex effects on daily rhythms, which will be further investigated in the future. However, it is worth noting that the timing of food intake on the test day might influence the results of the analyses of daily rhythm-related parameters. Although it has been shown in rodent models that peripheral concentrations of secreted substances and PBMC gene expression can be influenced by changes in feeding rhythm, rather than just by recent food intake[65,71], it is unclear at present which of these has a greater effect on the daily rhythms of secreted substances and PBMC clock gene expression in humans.

The present study had several limitations. Firstly, although it was a randomized trial, the participants could not be blinded to the intervention. Secondly, the people who applied to join the trial might already have been interested in TRF or wished to improve their health through making a dietary change, and most were women. Thirdly, the number of participants in the trial was relatively small and they may not have been representative of the wider population. Fourthly, the potential barriers to TRF were not analyzed. Fifthly, the participants in the TRF groups were required to consume their meals within an 8 h period, but the specific timing and duration of their meals varied within each group, which may have influenced the results. The influence of the duration of food consumption on the effects of TRF requires further investigation. Sixthly, the changes in the eating periods that were made in the TRF groups may have caused changes in the duration of fasting prior to testing, which might have influenced the results. Lastly, daily rhythm-related parameters were measured in limited numbers of participants and few time points were assessed. To better assess the effects of TRFs on daily rhythms, further, larger studies should be conducted that include shorter intervals between measurements and more than one diurnal cycle.

## Methods
**Study design**. We conducted a randomized, controlled trial, in which participants were randomized to an eTRF group (eating during no more than 8 h between 06:00 and 15:00 and fasting for the rest of the day), an mTRF group (eating during no more than 8 h between 11:00 and 20:00 and fasting for the rest of the day), and a control group (eating ad libitum over 8 h each day). Participants in the eTRF and mTRF groups were only allowed to consume water, flavored carbonated water, unsweetened tea, and coffee during the fasting period. The primary outcome was the change in HOMA-IR, an index of insulin resistance that is calculated using the fasting glucose and insulin concentrations. The secondary outcomes were changes in energy intake, fasting glucose, body mass, body composition, blood pressure, blood lipid concentrations, inflammatory markers, liver enzymes, immune cells, gut microbiota, sleep quality, and appetite. Change in daily rhythms of plasma adipokine concentrations and PBMC clock gene expression were measured as exploratory analyses.

**Study participants**. This clinical trial was conducted at Peking Union Medical College Hospital (PUMCH, China), approved by the hospital's ethics committee, and conducted according to the Helsinki Declaration of 1975. Prior to enrolling participants, the study was registered at chictr.org.cn (ChiCTR2000029797). Participants were recruited from the Beijing area from Feb. 16th, 2020, to Mar. 22nd, 2020, by means of posters, emails, flyers, social media, and website advertisements. Ninety participants who were in the habit of eating over more than 8 h per day and who did not have recent experience of fasting were recruited into the trial after providing their written informed consent.

**Diets and compliance**. Participants in the eTRF group were instructed to choose an 8 h eating period between 6:00 and 15:00 and to fast for the rest of the day. Those in mTRF group were instructed to choose an 8 h eating period between 11:00 and 20:00 and to fast for the rest of the day. Participants in the control group could eat ad libitum, following their usual eating regimens, with food being consumed over more than 8 h per day. The participants maintained their habitual alcohol intake during the trial, which was no more than twice a week, as required in the eligibility criteria. Alcohol intake was forbidden on the test days and the preceding days. To ensure compliance, participants were required to take photos of their food as they began to eat and as they finished and to send them privately to the investigators using a WeChat-supported web message-sending applet. All participants wrote a consent form and guaranteed to supply real data about food intake at the beginning of the trial. The investigators checked their posts every day and participants who failed to post those photos for more than 3 days, which meant they could not fulfill the required 90% completion rate, were considered to have failed to complete the trial. The energy content of each meal was estimated using China Food Composition Database[72]. One designated researcher who had got a good clinical practice certificate was trained to estimate the number of different types of food using the posted photos, which would be double-checked by another researcher. Standardized measurement guides were used to assess portion sizes. The records for all the meals of every participant were included in the analysis, except for non-compliant days. To estimate compliance, the number of person-days for each group was defined as 35 days (the length of the trial) multiplied by the number of participants who finished the trial. The compliance rate was calculated as the number of self-reported compliant days divided by the total number of person-days for each group. Because 28, 26, and 28 participants in the eTRF, mTRF, and control groups, respectively, completed the trial, the compliance levels were calculated to be 980 (35 × 28) for the eTRF group and 910 (35 × 26) for the mTRF group. Because participants were instructed to take either TRF regimen or normal diet regimen, they were not blinded to the assignment of the groups. Investigators who checked posted photos and estimated energy contents from photos were not blinded to the assignment of the group. Other investigators and statisticians were blinded during the study procedure, and were unblinded after all the data had been analyzed.

**Randomization procedure**. For the pilot RCT, participants were randomly assigned to either the eTRF, mTRF, or control group in a 1:1:1 ratio, using a computer-based random-number generator by designated researchers.

**Inclusion and exclusion criteria**. The inclusion criteria were: (1) 18–64 years old; (2) ability to attend the hospital at regular intervals; (3) ability to independently provide informed consent; (4) BMI between 17.5 and 30.0 kg/m²; (5) daily feeding period of more than 8 h; and (6) stable body mass (change < ±10% of current body mass during the 3 months prior to the study). The exclusion criteria were: (1) night-shift work more than once a week; (2) fasting during the preceding 8 weeks; (3) alcohol consumption more than twice a week; (4) pregnancy, gastrointestinal abnormalities or eating disorders, history of gastrointestinal surgery or systemic disease; (5) use of corticosteroid drugs, β-receptor blockers, or other drugs that might affect the findings; (6) a diagnosis of hypertension, diabetes, or other metabolic disease; and (7) a diagnosis of insomnia.

**Anthropometric measurements**. Body mass and percentage body fat were measured using an HBF-371 Bioelectrical impedance analyzer (Omron Healthcare Co., Kyoto, Japan). Height was measured using a metric tape, with the participant standing up straight against a wall. BMI was calculated using the body mass in kilograms divided by the height in meters, squared.

**Blood sampling and storage**. Blood sampling was performed at the beginning and the end of the trial. Fasting blood sampled were collected during the morning (07:00–08:30) after an overnight fast of at least 8 h. For those who participated in the analysis of daily rhythms, blood sampling was performed at 07:00, after an overnight fast, and at 12:00, 17:00, and 23:00. Plasma, serum, whole blood, and PBMC fractions were collected and either analyzed immediately or stored at −80 °C until assayed.

**Flow cytometric analysis**. PBMCs were separated from blood samples using Ficoll (GE Healthcare, Chicago, IL) and centrifugation. pTregs were counted by flow cytometry (FACS Canto plus, BD Biosciences, Franklin Lakes, NJ) using a fixed staining protocol of 5 µL antibody (Anti-human CD4 (RPA-T4) FITC, 11-0049-41; Anti-human CD25 (BC96) PE, 12-0259-41; Anti-human CD3 (UCHT1) APC, 17-0038-41; Anti-human CD127 (EBIORDR5), PERCP-CYAN, 45-1278-41; eBioscience, San Diego, CA) diluted in 100 µL PBS. Flow cytometric data were analyzed using FlowJo (Version 10.6.2, BD Biosciences) (Fig. S6).

**Fecal sample collection and storage**. Fecal samples were collected during the 3 days before the start of the trial and during the same period of time before the end of the trial. Detailed instructions regarding sample collection and transportation were provided by the study personnel and the participants were provided with containers with feces-preserving fluid. The participants were asked to collect approximately 2–3 g of feces using the spatula attached to the cover of the container, to place the fecal sample inside, and then to shake the container well. The containers were then delivered to investigators within 24 h and stored at −80 °C until the contents were analyzed.

**Biochemical measurements**. The plasma activities of AST, ALT, ALP, GGT, and lactate dehydrogenase; and the concentrations of LDL-C, HDL-C, total cholesterol, triglyceride, and glucose were measured using an automated analyzer (Beckmann-Coulter AU 5800, Brea, CA). Insulin was measured using an ADVIA Centaur XP (Siemens, Munich, Germany). Blood cells were analyzed using XN-2000 (Sysmex, Kobe, Japan). The IL-8 and TNF-α concentrations were measured using an Immulite 1000 (Siemens). The concentrations of resistin (AdipoGen Life Science, Liestal, Switzerland), leptin (Phoenix Pharmaceuticals, Burlingame, CA), and ghrelin (Thermo Fisher, Waltham, MA) were measured using ELISA kits on a microplate reader (Bio-Rad Laboratories).

**Real-time quantitative PCR**. RNA was pooled from PBMCs and used for cDNA synthesis. Transcript levels were then quantified by qPCR using SYBR qPCR mix (ABI-Invitrogen). The expression of each gene of interest was normalized to that of *ACTB* using the $2^{-\triangle\triangle CT}$ method. The primer sequences are listed in Table S3.

**Subjective sleep quality and eating habits**. The participants were required to maintain their normal sleeping habits throughout the trial and to avoid undergoing testing after a night shift. Sleep was analyzed using the PSQI questionnaire and eating habits were analyzed using the CNAQ.

**Analysis of the microbiota**. DNA was obtained from fecal samples using the QIAamp Fast DNA Stool Mini Kit (Qiagen), according to the manufacturer's protocol. The concentrations of the extracted DNA were measured using a Nanodrop, and DNA electrophoresis using a 1% agarose gel was performed to verify its integrity. To generate the 16S rDNA library, PCR analysis was performed using a 16S V3–V4 hypervariable region general primer set and a KAPA HiFi Hotstart ReadyMix PCR Kit (KAPA), and the PCR products were collected using an AxyPrep DNA gel extraction kit (Axygen). To establish qualified 16S rDNA libraries, the concentrations were measured using the Nanodrop, 1% agarose gel electrophoresis was performed, and quantitative testing with Qubit dsDNA HS Assay Kit was performed prior to sequencing. The 16S rDNA amplicon sequence results were analyzed using the Hiseq 2500 PE250 platform. The sequencing results were first assembled using PANDAseq 2.11 software to acquire clean reads[73]. An operational taxonomic units table was constructed using Usearch 10.0.259 and randomized leveling was performed on each sample to avoid sample size-related bias[74]. Alpha diversity, assessed using chao1, was analyzed using QIIME 2 2017.6.0[75], and the chao1 changes during the study were evaluated using one-way repeated-measures ANOVA, followed by the Holm-Sidak multiple comparisons test. Analyses of the changes in the gut microbial profiles were performed using LEfSe 1.0[76]. PICRUSt 1.0.0, based on closed-reference operational taxonomic units, was used to predict the abundances of functional categories, on the basis of COG analysis[75].

**Statistical analysis**. For the sample size calculation, we estimated that the eTRF group would show a 37.5% reduction in HOMA-IR and that the mTRF group would show no change in HOMA-IR during the 5-week trial[9]. The reported mean and standard deviation are 1.91 and 0.8 in Chinese people without obesity[77]. Therefore, to detect a 37.5% difference in HOMA-IR (1.91 × 0.375 = 0.7) between the TRF groups, the statistical power analysis indicated that 22 participants would have to complete the trial in each group in order to achieve 80% power (two-sided

test, $a = 0.05$), assuming a within-participants standard deviation of 0.8. Taking into consideration potential drop-outs during the trial, 30 participants were recruited for each group.

The energy that was consumed on non-compliant days was not included in the calculation of the mean daily energy consumption and the data for the eight participants who did not complete the trial were excluded from the analyses. Data are shown as Tukey box plots, with the mean value indicated by a line, unless otherwise stated. Data was collected using Microsoft Excel (Microsoft 365 MSO version 1904). Statistical calculations were performed using GraphPad Prism 7.0 (GraphPad software, La Jolla, CA). The D'Agostino & Pearson normality and Shapiro-Wilk tests were used to check the data distribution. To compare the three groups, one-way repeated-measures ANOVA was used, followed by the Holm-Sidak multiple comparisons test. To analyze the daily rhythms of circulating substance concentrations, the data at each time point were first collated as percentages of the mean of the values at all the time points, then analyzed using two-way repeated-measures ANOVA, with time of day and feeding regimen as the two independent variables. $P < 0.05$ was considered to represent statistical significance.

Cosinor analysis was applied to the clock gene expression data. Cosinor model curves were plotted using the clock gene expression data for each participant on each test day using the following function and the amplitude, MESOR, phase shift and acrophase (peak time) for each curve[78]:

$$f(t) = \text{Mesor} + \text{Amplitude} \times \cos\left(2\pi \times \frac{t}{24} + \text{Phaseshift}\right).$$

**Reporting summary**. Further information on research design is available in the Nature Research Reporting Summary linked to this article.

## Data availability

The individual de-identified participant microbiota metagenomic sequencing data can be accessed from the BioProject Database of National Centre for Biotechnology Information with the dataset accession number PRJNA786689. The daily rhythms-related source data underlying Fig. 4 and Supplementary Figs. 1, 2 are provided with this paper. The other individual de-identified participant data are not openly available due to participant confidentiality and will be shared by the corresponding author upon reasonable request for academic use. The study protocol is available as a supplementary file. China Food Composition Database was used in this manuscript[72]. Source data are provided with this paper.

## Code availability

No code was involved in this manuscript.

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

## Acknowledgements

This work was supported by grants from the CAMS Innovation Fund for Medical Sciences (CIFMS) (No.2016-I2M-1-001) (Y.M. and H.Y.). The study sponsors played no role in study design, conduct, data acquisition, analysis, manuscript preparation or the decision to submit the manuscript for publication. We thank Mark Cleasby, PhD from Liwen Bianji (Edanz) (www.liwenbianji.cn) for editing the language of a draft of this manuscript.

## Author contributions

Y.M. and H.Y. oversaw the design, regulatory compliance, execution, and data analyses in this study. Y.M., H.Y., and Z.X. designed the study. Z.X., Y.S., Y.Y., D.H., H.Zhang, Z.H., H.Zhao, and H.Y. recruited participants, collected data, and monitored participants compliance. All the authors contributed to data analyses. Z.X., Y.S., Y.M., and H.Y. wrote the manuscript. All the authors contributed to the composition and revision of the manuscript and gave final approval to its content.

## Competing interests

The authors declare no competing interests.

## Additional information

**Peer review information** *Nature Communications* thanks Heather Allore, Marta Garaulet and the other anonymous reviewer(s) for their contribution to the peer review this work. Peer reviewer reports are available.

