## [Peer Review File · Nature Communications]

Reviewers' Comments:

Reviewer #1:

Remarks to the Author:

Xie, et. al. present results on an intervention comparing early time-restricted feeding (eTRF) to midday time-restricted feeding (mTRF) for changes in weight loss, body composition, insulin resistance, energy intake, blood pressure, inflammation, and gut microbiota diversity in human adults without obesity. These results are the first to support a benefit of TRF early in the day compared to a midday eating period. The large sample size and three-group designs are both strengths and these data would be of high interest to readers.

Though the data are of the data are of great interest, too little detail in the methods, inappropriate between-group statistical tests, and interpretation of results that extend beyond the presented data are major flaws with this manuscript.

Major Comments:

1. Too little detail in the methods prevents a thorough evaluation of this manuscript. There are key details missing in nearly every section of the Methods, particularly the statistical methods:
 - a.) Thorough information on the statistical power analysis is not provided. What evidence was used to calculate the sample size? How were missing data handled? Were any data excluded and, if so, why?
 - b.) Information related to dietary intake is insufficient. How was the dietary data collected and extracted from the photos and what personnel analyzed these? What nutrient analysis software was used to analyze these data? How was data validity determined? How many participant records were included?
 - c.) What were the primary outcomes and were these preregistered?
2. Overall grammar and writing and the presentation of data (in tables and figures) needs improvement.
3. Adherence data were collected but are not reported and these data are critical for evaluating the data. A statement in the discussion insinuates that participants were highly adherent (Line 227) but this statement is not supported with the data included.
4. Inappropriate statistical methods were used to analyze and interpret the data. The methods state that ANOVAs were performed but Table 2 presents only within-group changes. Appropriate two-way, between-group tests were apparently not performed. It is not clear if adjustment for multiple comparisons was performed with a vast number of t-tests in the microbiota data.
5. Interpretation of results is extrapolated beyond what the data suggest.

Minor Comments:

Line 31 – Change to “improved fasting glucose”

Line 33 – “eTRF is superior to mTRF with regard to many aspects of metabolic health” – make statement less definitive as this is the first evidence of such.

In-text citations are not formatted appropriately.

Use person-first language in describing disease states (e.g. in Line 71 – change to “persons with obesity.”)

94 – change to “self-reported adherence”

Section starting at Line 99 – inadequate information related to energy intake assessment has been provided. How was the dietary data collected (dietary recall, diet records, etc.)? What software was used to analyze these data? How was data validity determined? How many participant records were included?

Line 120 – The first summary sentence is misleading and should be rephrased to appropriately describe the statistical method. You note that there are significant differences across groups but the only significant between-group difference was eTRF and control.

Line 332 – why was alcohol forbidden and why were participants who consumed alcohol more than once per week excluded? This could be a major contributing factor to the modest weight loss observed, particularly in the eTRF group.

Reviewer #2:

Remarks to the Author:

The authors present a 5-week randomized trial conducted at a single site to compare two TRFs in healthy, non-obese adults in China. There are several important findings. The manuscript would be strengthened by having improved English grammar throughout.

On line 69 could the authors clarify "eTRF (8h out of 06:00-15:00), mTRF (8h out of 11:00-20:00)"? Does this mean that the participants fasted for 8 hours of these windows or could eat during 8 hours of these windows?

As all participants were Asian, this is not needed in Table 1. There appears to be a higher % of women, did men not meet the inclusion? More details on the 312 that did not meet inclusion would be helpful.

The trial was powered for the HOMA-IR primary outcome. However, there are many secondary outcomes and these should be controlled for multiple comparisons as the familywise Type I error is inflated. Line 454's Holm-Sidak's is for within outcome comparison of the 3 groups, this is fine. But it is not controlled across outcomes.

Line 100, unclear how posted pictures were used to calculate energy.

Lines 193-195 these values are carried out to many digits, this seems unsupported given the reduced sample size.

Randomization section starting at line 341 is repeated starting at line 364.

The authors have not included any limitations, which should be revised. The sample does not appear to be representative of the population.

Heather Allore, PhD

Reviewer #3:

Remarks to the Author:

The novelty of this paper is that authors compare two different time windows for the time restricting feeding, one early eTRF (from 6 a.m to 3 p.m) and one late mTRF (from 11 a.m. to 8 p.m.). Authors found that eTRF was more effective than mTRF at improving insulin sensitivity. Furthermore, eTRF, but not mTRF, showed improvements on reducing fasting plasma glucose, reducing both body weight and body fat, alleviating inflammation, and increasing gut microbiota diversity.

I think it is a relevant question for the general population, TRF-based diets are reaching great popularity nowadays, and knowing the time window in which we should fast to achieve better results is relevant. However, there are several methodological aspects that should be addressed before finally accepting the work.

1) While authors report total energy intake in the control, eTRF and mTRF groups, no information is available in the timing of food intake. Furthermore, it is not known in the eTRF window, if the fasting duration is of 8 hours always, this would be the case if everybody had a breakfast at 6. A.m. If this is not the case the fasting duration would include night-fasting during sleep + 8 hours from 6 a.m. to 3 p.m. I suppose that the authors have this information, which is relevant to achieve a better understanding of the mechanisms implicated in the results and to design recommendations for the population. It is also relevant to know the timing of food intake in the control group.

2) Could authors better explain the conditions during the two test days (at baseline, and at the end of the treatment), there were differences in the timings for food intake, sleep duration etc, that may influence the results?

3) Blood sampling in those volunteers subjected to circadian rhythm parameters was performed at 7, after overnight fast, 12:00, 17:00 and 23:00. Although clock times were the same for all conditions, this was not the case for the circadian timing, or the timing related to behaviors, especially for the timing of food intake. These could be highly influencing the results in plasma serum, and even the PBMCs fractions. Particularly in adipokine concentrations. So, we don't know if the changes in the adipokines levels at specific timing of the day were due to fasting condition or the timing of the closest food for example at 12:00 we expect to be at least 6h in fasting in the eTRF, and only 1h of fasting in the mTRF. This is relevant to understand the differences and the mechanisms involved. Could authors explain better?

4) With respect to statistical analyses for clock gene assessments, why do authors use the cosinor analysis to determine amplitude, as I understand authors only determined 4 timing points, and

they were all during the day time (7, after overnight fast, 12:00, 17:00 and 23:00), I would like to see the figures to better assess if a cosine model fits well with the data. Furthermore, were all the rhythms significant? (could authors include this figure in supplemental data).

5) About fecal sample collection, authors indicate that samples were collected within three days before the start of the trial? I suppose that they were also collected after the trial in order to compare the effects of the different conditions on alpha diversity. When were the samples collected at the end?

6) For the Microbiota Analysis, which is the index used for alpha diversity? Authors do not show bacteria changes with the conditions, it would be interesting to see if different conditions result in differentiated microbiota profiles or changes in the function. Do authors have this information?

7) Finally, some authors have reported Barriers to TRF such as work schedules, family commitments and social events. Do authors have this information? If not, this should be included as a limitation.

This is a relevant aspect to consider before recommending TRF to the population.

8) Results

How do authors explain changes in resistin and ghrelin towards an increase at 12:00 and 23:00 respectively in the eTRF condition.

9) Conclusion

How can the authors conclude that the peripheral circadian rhythms activities were involved in the beneficial effects of both TRFs?

REVIEWER COMMENTS

Reviewer #1 (Remarks to the Author):

Xie, et. al. present results on an intervention comparing early time-restricted feeding (eTRF) to midday time-restricted feeding (mTRF) for changes in weight loss, body composition, insulin resistance, energy intake, blood pressure, inflammation, and gut microbiota diversity in human adults without obesity. These results are the first to support a benefit of TRF early in the day compared to a midday eating period. The large sample size and three-group designs are both strengths and these data would be of high interest to readers.

Though the data are of the data are of great interest, too little detail in the methods, inappropriate between-group statistical tests, and interpretation of results that extend beyond the presented data are major flaws with this manuscript.

We appreciate Reviewer #1 to confirm our merits in this manuscript and to expect that our data would be high interest to readers. Based on the specific comments on lack of details in methods and statistical tests, and on over-interpretations for some results, we have corrected and improved them already. Overall, we have thoroughly revised each of these specific aspects through rewriting the relative paragraphs that have been highlighted by yellow color.

Major Comments:

1. Too little detail in the methods prevents a thorough evaluation of this manuscript. There are key details missing in nearly every section of the Methods, particularly the statistical methods:

a.) Thorough information on the statistical power analysis is not provided. What evidence was used to calculate the sample size?

We appreciate Reviewer #1 for above comment and question. We have added the information on both statistical power analysis and sample size calculation into the Methods section of revised manuscript. As the evidence to be used for sample size

calculation, the change of HOMA-IR, which was the primary outcome of this trial, was reported to be -37.5% after TRF.¹ The reported mean and standard deviation were 1.91 and 0.8 in non-obese Chinese people². The change of HOMA-IR in control group was assumed to be 0. To detect 37.5% change in HOMA-IR between TRF groups and control group, a statistical power analysis indicated that 22 completers in each group can provide 80% power for a significant t test at 0.05 level, assuming a within-subjects standard deviation of 0.8. Considering drop-outs and P value adjustment for multiple tests, 30 participants in each group were targeted as the sample size.

How were missing data handled?

We apologize for not providing this information. In the revised manuscript, we have added the relative information of missing data into the Methods section. Here, we briefly introduce the added content: 82 among 90 participants completed the trial and 8 participants discontinued in the middle of the trial. Calories in non-adherent days were not reported by participants and they were not included in calculating the daily average calories. The final average calory for each participant was calculated as: calories sum in all adherent days divided by number of adherent days for each participant. The effect was minimal because non-adherent days were less than 10% for each participant who completed the trial. Furthermore, there is no other missing data than the calories in non-adherent days in this trial.

Were any data excluded and, if so, why?

We appreciate Reviewer #1 for this question. 8 participants discontinued in the middle of the trial. The data of those 8 participants who discontinued were excluded from analysis. We supplemented information related to the eight discontinued participants in the Methods section. As answered in the last question, the calories in non-adherent

days were not included for analysis, and all the other data of those 82 participants who completed the trial was included for analysis.

b.) Information related to dietary intake is insufficient. How was the dietary data collected and extracted from the photos and what personnel analyzed these?

We appreciate Reviewer #1 for above comment and questions. We apologize for not showing this information in the submitted manuscript. We have further clarified the relative information in the revised manuscript. Participants were instructed to take clear photos of their every meal during the trial. Designated researchers in our team were trained to estimate the number of different types of food, and they enquired participants if a photo was not good for food identification.

What nutrient analysis software was used to analyze these data?

We appreciate this question. We have added the relative information into the revised manuscript. The calories of every meal were calculated using a commercially available mobile App named Boohee.³ The records of all the meals of every participant who completed the trial were included for analysis except for non-adherent days.

How was data validity determined? How many participant records were included?

We have also added the relative information into the revised manuscript, based on this question. Briefly, all participants wrote a consent form and guaranteed to supply real data about food intake at the beginning of the trial. Besides, researchers checked about food intake information in the follow-up inquiries.

The records of all 82 participants who completed the trial were included for analysis except for the records of food intake data in non-adherent days.

c.) *What were the primary outcomes and were these preregistered?*

We have added this information in the revised manuscript. Briefly, the primary outcome was change in HOMA-IR (insulin resistance index calculated with fasting glucose and fasting insulin) and this was preregistered on chictr.org.cn (ChiCTR2000029797).

2. *Overall grammar and writing and the presentation of data (in tables and figures) needs improvement.*

We appreciate Reviewer #1 for this comment. We have been helped by the experienced peers under the background of English writing and speaking.

3. *Adherence data were collected but are not reported and these data are critical for evaluating the data. A statement in the discussion insinuates that participants were highly adherent (Line 227) but this statement is not supported with the data included.*

We appreciate Reviewer #1 for this comment. The adherence data was presented in the first paragraph of Results section in the previously submitted manuscript. We have supplemented relative information for better clarification. The total adherent person-days of each group was defined as the multiplication product of 35 days (5 weeks, the length of the trial) and the number of participants who completed the trial and were used for data analysis. Because 28, 26 and 28 participants from eTRF, mTRF and control group completed the trial, the adherent person-days were calculated to be 980 ($35 \times 28 = 980$) in eTRF group, 910 ($35 \times 26 = 910$) in mTRF group. During this trial, 31 person-days in eTRF group and 16 person-days from mTRF group reported non-adherence. As a result, the self-reported adherence to the regimens were 949 out of 980 (96.8%) in eTRF group, and 894 out of 910 (98.2%) in mTRF group.

4. Inappropriate statistical methods were used to analyze and interpret the data. The methods state that ANOVAs were performed but Table 2 presents only within-group changes. Appropriate two-way, between-group tests were apparently not performed.

We appreciate Reviewer #1 for above comments. The “within-group” changes in Table 2 were actually meant to show p-values among three groups in ANOVA analysis. The between-group tests were performed with ANOVA analysis, as stated in Methods section. We apologize for not showing all the relative results in the submitted manuscript. We have corrected this error in revised manuscript and complemented all the results from ANOVA analysis in the supplemented files.

It is not clear if adjustment for multiple comparisons was performed with a vast number of t-tests in the microbiota data.

We appreciate this comment. Accordingly, we have improved the relative descriptions in the revised manuscript. The microbiota data was compared with one-way ANOVA followed by Holm-Sidak's multiple comparisons test. The adjustment for multiple comparisons was performed.

5. Interpretation of results is extrapolated beyond what the data suggest.

We appreciated Reviewer #1 for this comment. We agree that some interpretations had such problem. According to this comment, we have thoroughly revised the sentences that may cause confusion or over-interpretations in interpretation based on the comments.

Minor Comments:

Line 31 – Change to “improved fasting glucose”

We appreciate Reviewer #1 for this comment, and we have corrected this error in the revised manuscript.

Line 33 – “eTRF is superior to mTRF with regard to many aspects of metabolic health” – make statement less definitive as this is the first evidence of such.

We appreciate Reviewer #1 for this comment, and we have corrected this error in the revised manuscript.

In-text citations are not formatted appropriately.

Use person-first language in describing disease states (e.g. in Line 71 – change to “persons with obesity.”)

94 – change to “self-reported adherence”

We appreciate Reviewer #1 for careful readings and rigorous judgements. We have corrected all of above errors in the revised manuscript.

Section starting at Line 99 – inadequate information related to energy intake assessment has been provided. How was the dietary data collected (dietary recall, diet records, etc.)? What software was used to analyze these data? How was data validity determined? How many participant records were included?

We appreciate Reviewer #1 for these comments. We have answered these questions previously in the Question b.

Line 120 – The first summary sentence is misleading and should be rephrased to appropriately describe the statistical method. You note that there are significant differences across groups but the only significant between-group difference was eTRF and control.

We appreciate Reviewer #1 for this comment. We have rephrased this sentence and improved the description of statistical result.

Line 332 – why was alcohol forbidden and why were participants who consumed alcohol more than once per week excluded? This could be a major contributing factor to the modest weight loss observed, particularly in the eTRF group.

We appreciate Reviewer #1 for this comment. We apologize for making this writing mistake in originally submitted manuscript. It has been corrected in the revised manuscript. In fact, we requested participants to maintain their routine alcohol intake during the trial. Alcohol was only forbidden one day before test days. As specified in the eligible criteria, participants were not allowed to take alcohol more than twice a week.

Reviewer #2 (Remarks to the Author):

The authors present a 5-week randomized trial conducted at a single site to compare two TRFs in healthy, non-obese adults in China. There are several important findings. the manuscript would be strengthened by having improved English grammar throughout.

We appreciate Reviewer #2 for acknowledging our findings to be important in this research area. We admit that English grammar of our original manuscript should be improved. Accordingly, we have invited researcher of native English speaker to finally edit and proof-read the revised manuscript.

On line 69 could the authors clarify "eTRF (8h out of 06:00-15:00), mTRF (8h out of 11:00-20:00)"? Does this mean that the participants fasted for 8 hours of these windows or could eat during 8 hours of these windows?

We apologize that this meaning was not expressed clearly in the original manuscript. We have improved the relative sentence in the revised manuscript. Briefly, our intended to express that: eTRF requested participants to eat less than 8 hours in the time window of 06:00 to 15:00, while mTRF requested participants to eat less than 8 hours in the time window of 11:00 to 20:00.

As all participants were Asian, this is not needed in Table 1. There appears to be a higher % of women, did men not meet the inclusion? More details on the men that did not meet inclusion would be helpful.

We appreciate Reviewer #2 for above suggestion and questions. We have modified Table 1 accordingly. We also appreciate Reviewer #2 for the comment on the disproportion of women and men included in this trial. Participants in this trial was recruited from local via poster, social media, etc. Briefly, It was suggested that women were more interested in TRF than men as more women applied. There was no significant difference in sex distribution among the three groups. We have supplemented this as new information in the Limited section of revised manuscript.

The trial was powered for the HOMA-IR primary outcome. However, there are many secondary outcomes and these should be controlled for multiple comparisons as the familywise Type I error is inflated. Line 454's Holm-Sidak's is for within outcome comparison of the 3 groups, this is fine. But it is not controlled across outcomes.

We appreciate Reviewer #2 for this comment. Since there were multiple secondary outcomes to be tested, adjustment on P value among secondary outcomes and including primary outcomes could reduce the power significantly. Thus, no P value adjustment was applied in testing secondary outcomes. Instead, we decided to treat all secondary outcomes as exploratory to address the concerns in multiplicity. This was also the typical design in the previously published TRF studies with a specific primary outcome.⁴⁻⁶ When comparing every outcome among three groups, the

Holm-Sidak's multiple comparisons test was applied after a one-way repeated measure ANOVA analysis.

Line 100, unclear how posted pictures were used to calculate energy.

We appreciate Reviewer #2 for this comment. We apologize for not showing this information in the submitted manuscript. We have further clarified the relative information in the Method section of revised manuscript. Participants were instructed to take clear photos of their every meal during the trial. Designated researchers in our team were trained to estimate the number of different types of food, and they enquired participants if a photo was not good for food identification. The calories of every meal were calculated using a commercially available mobile App named Boohee.³ The records of all the meals of every participant who completed the trial were included for analysis except for non-adherent days.

Lines 193-195 these values are carried out too many digits, this seems unsupported given the reduced sample size.

We appreciate Reviewer #2 for this comment. Mistakes on significant digits have been corrected in the revised manuscript.

Randomization section starting at line 341 is repeated starting at line 364.

We appreciate Reviewer #2 for this comment. This error has been corrected in the revised manuscript.

The authors have not included any limitations, which should be revised. The sample does not appear to be representative of the population.

We appreciate Reviewer #2 for this suggestion. We admit this trial had several limitations. According to this suggestion, we have added both a limitation part and complemented limitations of this trial in the revised manuscript.

Reviewer #3 (Remarks to the Author):

The novelty of this paper is that authors compare two different time windows for the time restricting feeding, one early eTRF (from 6 a.m to 3 p.m) and one late mTRF (from 11 a.m. to 8 p.m.). Authors found that eTRF was more effective than mTRF at improving insulin sensitivity. Furthermore, eTRF, but not mTRF, showed improvements on reducing fasting plasma glucose, reducing both body weight and body fat, alleviating inflammation, and increasing gut microbiota diversity.

I think it is a relevant question for the general population, TRF-based diets are reaching great popularity nowadays, and knowing the time window in which we should fast to achieve better results is relevant. However, there are several methodological aspects that should be addressed before finally accepting the work.

We appreciate Reviewer #3 for acknowledging the novelty of our study and acknowledging our results to reach relevant for general population. Especially, we also admit that several methodological aspects should be addressed according to Reviewer #3 comment. We have put all of these relative modifications in the revised manuscript.

While authors report total energy intake in the control, eTRF and mTRF groups, no information is available in the timing of food intake. Furthermore, it is not known in the eTRF window, if the fasting duration is of 8 hours always, this would be the case if everybody had a breakfast at 6. A.m. If this is not the case the fasting duration would include night-fasting during sleep + 8 hours from 6 a.m. to 3 p.m. I suppose that the authors have this information, which is relevant to achieve a better understanding of the mechanisms implicated in the results and to design recommendations for the population. It is also relevant to know the timing of food intake in the control group.

We appreciate Reviewer #3 for above comments. Based on these comments, we have added the mentioned information into the Limitations section of revised manuscript. Firstly, we have further clarified that this trial requested participants in TRF groups to eat for less than 8 hours and fast for 16 hours in a day. The eating duration was not of 8 hours always, because participants were thought to meet the requests as long as they restricted their eating period less than 8 hours in specific time window. On the other hand, our aim of this trial was to find the different effects of TRFs with different eating windows, rather than to compare TRFs with different fasting durations in a specific window. Thus, the specific food intake timing and specific fasting duration were not analyzed for each participant. We agree that the different fasting durations might have influenced the effects of TRFs, but our results were still because the focus of this trial was the time window of TRF instead of the fasting duration of TRF. It is worth noting that a recent trial carried out by Krista Varady et al. also studied the difference between TRF with 4-hour eating period and TRF with 6-hour eating period.⁶

Could authors better explain the conditions during the two test days (at baseline, and at the end of the treatment), there were differences in the timings for food intake, sleep duration etc, that may influence the results?

We appreciate Reviewer #3 for above questions. They are helpful for us to realize the weakness of relative descriptions in the original manuscript. Accordingly, we have improved the revised manuscript. Participants were required to maintain the same sleeping pattern over the trial. The change in eating period in TRF groups inevitably resulted in different pre-testing fasting durations between baseline and end dates or between different groups. However, it was reported that fasting durations longer than 6 hours did not further influence testing results of the key parameters tested in this trial significantly.^{7,8} In this trial, pre-testing fasting durations in all conditions were requested to be no less than 8 hours to reduce their impact on testing results.

Remarkably, the similar situation with different pre-testing fasting durations also happened in previous studies.⁹ The influences of food intake timings between baseline and end dates were difficult to be evaluated because participants were allowed to eat ad libitum in restricted eating periods. We have included the potential influences of different pre-testing fasting durations between different conditions in the Limitation section of the revised manuscript. The requirements of minimal fasting duration were also further clarified in the Methods section of revised manuscript.

Blood sampling in those volunteers subjected to circadian rhythm parameters was performed at 7, after overnight fast, 12:00, 17:00 and 23:00. Although clock times were the same for all conditions, this was not the case for the circadian timing, or the timing related to behaviors, especially for the timing of food intake. These could be highly influencing the results in plasma serum, and even the PBMCs fractions. Particularly in adipokine concentrations. So, we don't know if the changes in the adipokines levels at specific timing of the day were due to fasting condition or the timing of the closest food for example at 12:00 we expect to be at least 6h in fasting in the eTRF, and only 1h of fasting in the mTRF. This is relevant to understand the differences and the mechanisms involved. Could authors explain better?

We appreciate Reviewer #3 for these comments. They are very helpful for us to improve our interpretations on results. The corresponding discussions have been added into the Discussion section of revised manuscript.

Here, we briefly introduce the newly added contents. It was reported in rats that the circadian rhythm of some adipokines was related to feeding rhythms, but not the closest food intake.¹⁰ For PBMC clock genes expressions, food intake was reported in mice to show influence on PBMC gene expressions after 20 hours but not after 4 hours.¹¹ Therefore, the changes in PBMC gene expressions could be caused by the feeding rhythm in our trial, but not the closest food intake. However, whether the closest food intake showed more influence on the rhythm of adipokines and PBMC

clock genes expressions in human remains to be further elucidated in our future studies.

With respect to statistical analyses for clock gene assessments, why do authors use the cosinor analysis to determine amplitude, as I understand authors only determined 4 timing points, and they were all during the day time (7, after overnight fast, 12:00, 17:00 and 23:00), I would like to see the figures to better assess if a cosine model fits well with the data. Furthermore, were all the rhythms significant? (could authors include this figure in supplemental data).

We appreciate Reviewer #3 for the comments. They help us analyze data more carefully. We have revised the Results and Discussion sections accordingly. It was reported that clock genes in PBMC experienced circadian rhythm and the Cosinor model was able to well reflect the rhythm changes of clock gene expressions¹². In our study, the circadian rhythm was indeed assessed in samples only collected during four daytime points because participant refused to receive testing late at night. Although the majority of our data fitted well with Cosinor models, some data did not. The fitting curves and corresponding R-squared values have been added in supplemented files. The parameters that were presented were only for exploratory purpose. Statistical tests were withdrawn. We have discussed these in the Limitations section of revised manuscript.

About fecal sample collection, authors indicate that samples were collected within three days before the start of the trial? I suppose that they were also collected after the trial in order to compare the effects of the different conditions on alpha diversity. When were the samples collected at the end?

We appreciate Reviewer #3 for above comment and questions. Same as what were mentioned by Reviewer 3, the fecal samples collected at two time points: before the start of the trial, as well as prior to the end of the trial. These fecal samples collected

from two time points were compared on alpha diversity. Together, all of the relative information has been added into the revised manuscript.

For the Microbiota Analysis, which is the index used for alpha diversity?

The alpha diversity index used in this trial was chao1 that is a qualitatively measure of alpha diversity. This information has been added into the revised manuscript.

Authors do not show bacteria changes with the conditions, it would be interesting to see if different conditions result in in differentiated microbiota profiles or changes in the function. Do authors have this information?

We appreciate Reviewer #3 for above comments. Based on Reviewer #3's comments, the relative information has been added in the Results section of revised manuscript. No significant differences were found in relative microbiota abundances between baseline and follow-up in eTRF and control groups. In mTRF group, the relative abundances of *Escherichia_Shigella* and *Weissella* were enriched in baseline at genus level, and the relative abundance of *Leuconostocaceae* was enriched in follow-up at family level. PICRUSt was used to analyze functional genes of microbial communities in each group. Using function predictions based on clusters of orthologous group (COG) analysis, we found 29, 26, 1 significantly different functional COGs between baseline and follow-up testing results in eTRF group, mTRF group and control group, respectively.

Finally, some authors have reported Barriers to TRF such as work schedules, family commitments and social events. Do authors have this information? If not, this should be included as a limitation. This is a relevant aspect to consider before recommending TRF to the population.

We appreciate Reviewer #3 for above comments. We understand the above-mentioned information in previous reports. In our current study, participants from all three groups reported some non-adherent days during the trial suggesting barriers to TRF. However, some participants refused to inform us the particular reason of non-adherence in some cases for their privacy concerns. Therefore, it was not possible for us to completely collect and analyze the whole information on the barriers in this trial. As recommended by Reviewer #3, we have included this as a limitation in the revised manuscript.

Results

How do authors explain changes in resistin and ghrelin towards an increase at 12:00 and 23:00 respectively in the eTRF condition.

We appreciate Reviewer #3 for this comment. Based on this comment, we have realized the relative weaknesses in our original description. Thus, some new discussions have been added into the Discussion section of revised manuscript.

Here, we briefly introduce the newly added information. The rhythm of resistin was reported to be related to the feeding rhythm in rats¹⁰, and TRF was reported to influence the circulating level of resistin in men.¹³ The impact of change in feeding window on resistin rhythm had not been reported before. The change in resistin rhythm might merely be a reaction to the change of feeding rhythm.

The rhythm of ghrelin was reported to be synchronized to TRF in mice, which showed an increase before feeding period.¹⁴ It should be noted that ghrelin was reported to be mainly accorded to sense of hunger.¹⁵ Therefore, the higher level of ghrelin at 23:00 in eTRF group than that in mTRF group might be partially due to a longer fasting duration in eTRF group when reaching that time point.

Conclusion

How can the authors conclude that the peripheral circadian rhythms activities were involved in the beneficial effects of both TRFs?

We appreciate Reviewer #3 for above question. In fact, we did not decide to make this conclusion. After knowing your question, we have found our weakness in the relative description and have rephrased our sentences to make Conclusions less confusing. Indeed, we tried to express that both TRFs seemed to influence peripheral circadian rhythm, and that the different effects of two TRFs might be caused by their different effects on peripheral circadian rhythm.

References

1. Gabel, K. *et al.* Effects of 8-hour time restricted feeding on body weight and metabolic disease risk factors in obese adults: A pilot study. *Nutrition and healthy aging* vol. 4 345–353 (2018).
2. Zhou, Z. wei *et al.* Serum fetuin-A levels in obese and non-obese subjects with and without type 2 diabetes mellitus. *Clin. Chim. Acta* **476**, 98–102 (2018).
3. Zou, H. *et al.* Effect of caloric restriction on BMI, gut microbiota, and blood amino acid levels in non-obese adults. *Nutrients* **12**, 1–15 (2020).
4. Sutton, E. F. *et al.* Early Time-Restricted Feeding Improves Insulin Sensitivity, Blood Pressure, and Oxidative Stress Even without Weight Loss in Men with Prediabetes. *Cell metabolism* vol. 27 1212-1221.e3 (2018).
5. Wilkinson, M. J. *et al.* Ten-Hour Time-Restricted Eating Reduces Weight, Blood Pressure, and Atherogenic Lipids in Patients with Metabolic Syndrome. *Cell Metab.* **31**, 92-104.e5 (2020).
6. Cienfuegos, S. *et al.* Effects of 4- and 6-h Time-Restricted Feeding on Weight and Cardiometabolic Health: A Randomized Controlled Trial in Adults with Obesity. *Cell Metab.* **32**, 1–13 (2020).
7. Hancox, R. J. & Landhuis, C. E. Correlation between measures of insulin resistance in fasting and non-fasting blood. *Diabetol. Metab. Syndr.* **3**, 23 (2011).
8. Clemmensen, K. K. B. *et al.* Role of fasting duration and weekday in incretin and glucose regulation. *Endocr. Connect.* **9**, 279–288 (2020).
9. Sutton, E. F. *et al.* Early Time-Restricted Feeding Improves Insulin Sensitivity, Blood Pressure, and Oxidative Stress Even without Weight Loss in Men with Prediabetes. *Cell Metab.* **27**, 1212-1221.e3 (2018).
10. Oliver, P. *et al.* Resistin as a putative modulator of insulin action in the daily feeding/fasting rhythm. *Pflugers Arch. Eur. J. Physiol.* **452**, 260–267 (2006).
11. Jordan, S. *et al.* Dietary Intake Regulates the Circulating Inflammatory Monocyte Pool. *Cell* **178**, 1102-1114.e17 (2019).

12. Wehrens, S. M. T. *et al.* Meal Timing Regulates the Human Circadian System. *Current biology : CB* vol. 27 1768-1775.e3 (2017).
13. Alam, I. *et al.* Recurrent circadian fasting (RCF) improves blood pressure, biomarkers of cardiometabolic risk and regulates inflammation in men. *J. Transl. Med.* **17**, (2019).
14. Yasumoto, Y. *et al.* Short-term feeding at the wrong time is sufficient to desynchronize peripheral clocks and induce obesity with hyperphagia, physical inactivity and metabolic disorders in mice. *Metabolism.* **65**, 714–727 (2016).
15. Gualillo, O., Lago, F., Gómez-Reino, J., Casanueva, F. F. & Dieguez, C. Ghrelin, a widespread hormone: Insights into molecular and cellular regulation of its expression and mechanism of action. *FEBS Lett.* **552**, 105–109 (2003).

Reviewers' Comments:

Reviewer #1:

Remarks to the Author:

Comments to authors:

1. Citations are not formatted correctly. See style guide and author instructions for formatting guidelines.
2. Line 57 – “less marked” is unclear. Consider changing “to have less marked effects” to “late TRF does not produce the same magnitude of improvements as either eTRF or mTRF.”
3. Line 57 – Consider adding results from 4-hr, late day TRF in Cienfuegos, 2020: 10.1016/j.cmet.2020.06.018
4. Line 58 – This is a broad statement that may not be supported by current literature; further, these are inappropriate citations for the claim. The references include one mice trial and two narrative reviews which allude to the possibility of circadian mechanisms mediating the benefits of the timing of eating but they do not provide data to directly support this hypothesis. Include RCT data from mice and humans (if there are such data) and specify which you are referring to.
5. Section starting on line 88 – there is still insufficient detail regarding energy intake:
 - a. What instructions were provided for participants to take photos? How were foods with a similar appearance but widely varying nutrient content assessed? For example, how was the fat content of dairy products (0%, 2%, 4%, etc.) or oil used for cooking determined by a photograph?
 - b. Were standardized measurement guides used to assess portion sizes?
 - c. What were the credentials of those who entered the data and how was interrater variability assessed?
 - d. What database does Boohee use to analyze the nutrient content?
 - e. What was the total energy intake at both time periods and how was the validity of records assessed?
6. Put body weight and fat loss results in context – a fraction of a kg is not a clinically relevant difference between groups. Further, overall weight loss was modest (though not out of line with prior studies on TRF).
7. Paragraph starting line 246 – please put the blood pressure results in context given prior literature. Why do you think your results are in contrast to prior findings?
8. Table 1 – consider including % after the number of female participants and excluding the line for males, as this is inferred.

Reviewer #2:

Remarks to the Author:

The authors have greatly revised the manuscript in light of the reviewers' comments. This has improved the clarity and add details needed to understand the trial. Thank you for your efforts to revise the manuscript. My previous comments have all be satisfactorily addressed and this has resulted in a stronger manuscript.

Reviewer #3:

Remarks to the Author:

the authors have addressed most of the points and recognized the limitations of the study. However, before finally accepting the manuscript authors should substitute the term circadian rhythms by daily rhythms, due to:

- 1) these rhythms may not be endogenous, and not being driven by the internal clock but by the behaviors
- 2) these rhythms do not approach to a cosinor
- 3) there is not data from the 24h only daytime data

When this is fixed the manuscript can be accepted

Reviewer #1 (Remarks to the Author):

Comments to authors:

1. Citations are not formatted correctly. See style guide and author instructions for formatting guidelines.

We appreciate Reviewer #1 for this comment. We have revised the citations in the revised manuscript according to instructions for formatting guidelines.

2. Line 57 – “less marked” is unclear. Consider changing “to have less marked effects” to “late TRF does not produce the same magnitude of improvements as either eTRF or mTRF.”

We appreciate Reviewer #1 for this comment. We have revised the relative sentence in the revised manuscript accordingly.

3. Line 57 – Consider adding results from 4-hr, late day TRF in Cienfuegos, 2020: 10.1016/j.cmet.2020.06.018

We appreciate Reviewer #1 for this comment. We have added the relative results in the revised manuscript, accordingly.

4. Line 58 – This is a broad statement that may not be supported by current literature; further, these are inappropriate citations for the claim. The references include one mice trial and two narrative reviews which allude to the possibility of circadian mechanisms mediating the benefits of the timing of eating but they do not provide data to directly support this hypothesis. Include RCT data from mice and humans (if there are such data) and specify which you are referring to.

We appreciate Reviewer #1 for above comments. We have included RCT data from mice studies and one human trial which all implicated associations between TRFs with different eating windows and rhythmic variations in the revised manuscript. The trials from mice or humans have been specified in the revised manuscript.

5. Section starting on line 88 – there is still insufficient detail regarding energy intake:
a. What instructions were provided for participants to take photos? How were foods with a similar appearance but widely varying nutrient content assessed? For example, how was the fat content of dairy products (0%, 2%, 4%, etc.) or oil used for cooking determined by a photograph?

We appreciate Reviewer #1 for above comments and questions. Participants were instructed by researchers to take photos of their food at the beginning of the trial.

Briefly, they need to take a picture of their whole meal, then pictures of each food were also taken. Pictures of foods with similar appearance were required to include the ingredients list if possible, and participants would be further enquired if researchers wondered about the type of foods. We admit that this method was only a close estimation of the real energy intake, but we presumed results using this method to be a better estimation than food diaries or food frequency questionnaires which were usually used in previous trials. Besides, the energy intakes were not required in each group. We have replaced the word 'calculated' with 'estimated' in the Energy intake part of Result section.

b. Were standardized measurement guides used to assess portion sizes?

We appreciate Reviewer #1 for this question. Standardized measurement guides were used to assess portion sizes. This information has been supplemented in the Methods section of the revised manuscript.

c. What were the credentials of those who entered the data and how was interrater variability assessed?

We appreciate Reviewer #1 for above questions. The researcher who entered the data had got a GCP (good clinical practice) certificate before the trial. Only one trained researcher entered the data during the trial, so no interrater variability was concerned in this trial. The entered data would be double-checked by another researcher who also got a GCP certificate. Relative information has been added in the Methods section of revised manuscript.

d. What database does Boohee use to analyze the nutrient content?

We appreciate Reviewer #1 for this question. Staff from Boohee app informed us that they used the China Food Composition Database in their app to analyze the nutrient content. [ref 1] However, no notification about this information was shown in their app or website, nor were they willing to send us a formal declaration about this information. As a result, we used data from China Food Composition Database to recalculate the energy intake data and got the same results. We have changed the method for calculating energy content to "China Food Composition Database" in the Methods section of the revised manuscript.

References:

[1] Yang, Y. X. ., Wang, G. & Pan, X. . *China Food Composition*. (Peking University Medical Press, 2009).

e. What was the total energy intake at both time periods and how was the validity of

records assessed?

We appreciate Reviewer #1 for above questions. Changes in energy intake were compared among three groups, and as a result, the total energy intake were not shown in previous manuscript. The average energy intake each day of participants in eTRF group during eTRF period were 1456 ± 274 kcal, that in mTRF group during mTRF period was 1537 ± 266 kcal. As to the validity of records, all participants wrote a consent form and guaranteed to supply real data about food intake at the beginning of the trial. Besides, researchers checked about food intake information every day and in the follow-up inquiries. This information has been supplemented in the Methods section of the revised manuscript.

6. Put body weight and fat loss results in context – a fraction of a kg is not a clinically relevant difference between groups. Further, overall weight loss was modest (though not out of line with prior studies on TRF).

We appreciate Reviewer #1 for above comments. We apologize for not clarifying these points, and relative explanations have been added in the Discussions section of the revised manuscript. Briefly, metabolic disorders can involve fat deposition, and a reduction in fat mass and percentage body fat may indicate an improvement in fat deposition, thus an improvement in metabolic health. However, the proof of this effect requires further visceral fat measuring parameters in future trials.

The relatively modest weight loss compared with prior studies may be the result of different participants inclusion criteria, with normal-weighted humans included in this trial, while mostly overweight or obese participants were included in prior eTRF studies.

7. Paragraph starting line 246 – please put the blood pressure results in context given prior literature. Why do you think your results are in contrast to prior findings?

We appreciate Reviewer #1 for above comment and question. We have rephrased relative paragraph to better clarify our points of views. Briefly, only one trial by Courtney Peterson et al. has evaluated the effects of eTRF on blood pressure which showed markedly reduction on blood pressure of participants. The result of blood pressure in eTRF group in the present trial was different from the trial by Courtney Peterson et al. On the other hand, the blood pressure results of the mTRF group in the present trial were in consistent with most previously published trial on mTRF, and the only one showing a reduction on blood pressure was assumed to be an “add-on” effects of anti-hypertensive drugs.

8. Table 1 – consider including % after the number of female participants and

excluding the line for males, as this is inferred.

We appreciate Reviewer #1 for this comment. Accordingly, we have modified Table 1 in the revised manuscript.

Reviewer #2 (Remarks to the Author):

The authors have greatly revised the manuscript in light of the reviewers' comments. This has improved the clarity and add details needed to understand the trial. Thank you for your efforts to revise the manuscript. My previous comments have all be satisfactorily addressed and this has resulted in a stronger manuscript.

We appreciate Reviewer #2 for acknowledging our revision to have addressed the comments and resulted in a stronger manuscript.

Reviewer #3 (Remarks to the Author):

the authors have addressed most of the points and recognized the limitations of the study.

We appreciate Reviewer #3 for acknowledging our revision to have addressed most of the points and recognized the limitations.

However, before finally accepting the manuscript authors should substitute the term circadian rhythms by daily rhythms, due to:

1) these rhythms may not be endogenous, and not being driven by the internal clock but by the behaviors

2) these rhythms do not approach to a cosinor

3) there is not data from the 24h only daytime data

When this is fixed the manuscript can be accepted

We appreciate Reviewer #3 for above comments. Based on these comments, we have substituted the term circadian rhythms by daily rhythms in our revised manuscript.

Reviewers' Comments:

Reviewer #1:

Remarks to the Author:

As previously stated, the results from this trial will be of great interest to the readership and thank you for this work. The authors have revised the methodology for dietary intake and have appropriately phrased the corresponding results. Other revisions have adequately addressed reviewer concerns.

Reviewer #1 (Remarks to the Author):

As previously stated, the results from this trial will be of great interest to the readership and thank you for this work. The authors have revised the methodology for dietary intake and have appropriately phrased the corresponding results. Other revisions have adequately addressed reviewer concerns.

We appreciate Reviewer #1 for acknowledging our revision to have addressed the concerns.